# A genetic circuit on a single DNA molecule as an autonomous dissipative nanodevice

Ferdinand Greiss [1] ✉, Nicolas Lardon[2], Leonie Schütz[3], Yoav Barak[4], Shirley S. Daube [1], Elmar Weinhold[3], Vincent Noireaux [5] & Roy Bar-Ziv [1] ✉

Realizing genetic circuits on single DNA molecules as self-encoded dissipative nanodevices is a major step toward miniaturization of autonomous biological systems. A circuit operating on a single DNA implies that genetically encoded proteins localize during coupled transcription-translation to DNA, but a single-molecule measurement demonstrating this has remained a challenge. Here, we use a genetically encoded fluorescent reporter system with improved temporal resolution and observe the synthesis of individual proteins tethered to a DNA molecule by transient complexes of RNA polymerase, messenger RNA, and ribosome. Against expectations in dilute cell-free conditions where equilibrium considerations favor dispersion, these nascent proteins linger long enough to regulate cascaded reactions on the same DNA. We rationally design a pulsatile genetic circuit by encoding an activator and repressor in feedback on the same DNA molecule. Driven by the local synthesis of only several proteins per hour and gene, the circuit dynamics exhibit enhanced variability between individual DNA molecules, and fluctuations with a broad power spectrum. Our results demonstrate that co-expressional localization, as a nonequilibrium process, facilitates single-DNA genetic circuits as dissipative nanodevices, with implications for nanobiotechnology applications and artificial cell design.

DNA nanotechnology seeks to construct nanodevices that perform mechanical work such as rotors and tweezers[1–3], or process information with architectures such as logic gates[4] and neural networks[5,6]. Most nanodevices rely on external nucleic-acid inputs to switch between states[3]. Dissipative DNA nanotechnology marks a shift to computing systems that could reversibly transition between states through a constant turnover of DNA or RNA strands, leading to pulsatile and oscillatory behavior in bulk solution[7,8]. Combining protein synthesis with DNA nanotechnology would advance this field toward single-molecule computation, as recently demonstrated by a device with logic gates using an RNA polymerase (RNAP) and gene co-localized on a DNA origami chip[9]. The integration of genetic circuits

designed for bulk reactions[10–15] would further extend their complexity at the miniaturization limit of a single DNA molecule toward autonomous and self-encoded dissipative systems.

A genetic circuit would operate as a single-DNA nanodevice if nascent regulatory proteins interact with the DNA on which they are encoded. This is unlikely to occur in dilute cell-free reactions, where entropy favors dispersion unless nascent proteins remain linked to the DNA by the transient complex of RNAP, messenger RNA (mRNA), and ribosome (Fig. 1a)[16–20]. Such a nonequilibrium mechanism of gene regulation has recently been suggested to explain how a decision-making circuit operates in dilute conditions[21]. It has been invoked in prokaryotic cells[22], where transcription and translation

[1]Department of Chemical and Biological Physics, Weizmann Institute of Science, Rehovot 7610001, Israel. [2]Department of Chemical Biology, Max Planck Institute for Medical Research, 69120 Heidelberg, Germany. [3]Institute of Organic Chemistry, RWTH Aachen University, 52056 Aachen, Germany. [4]Department of Chemical Research Support, Weizmann Institute of Science, Rehovot 7610001, Israel. [5]School of Physics and Astronomy, University of Minnesota, Minneapolis, MN 55455, USA. ✉e-mail: ferdinand.greiss@gmail.com; roy.bar-ziv@weizmann.ac.il

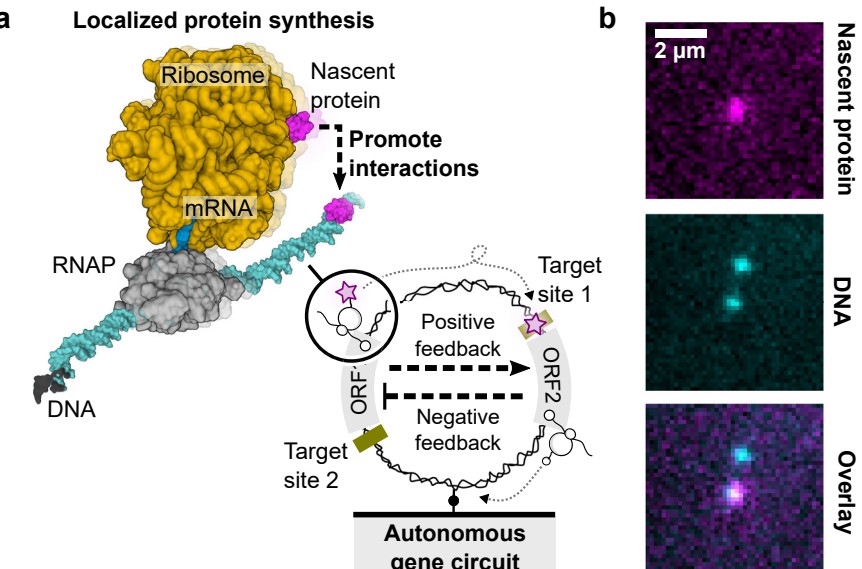

**a** Localized protein synthesis

**b**

Fig. 1 | **Genetic circuit as dissipative nanodevice driven by localized protein synthesis on a single DNA molecule. a** Interaction between the nascent protein and its target site on the same DNA is promoted by the transient RNAP-mRNA-ribosome complex (PDB: 6X9Q). Schematic of the coupled transcription (RNAP) and translation (ribosome) machines decoding open-reading frames (ORF) to synthesize and tether nascent proteins to the DNA, driving autonomous genetic circuits on the same DNA molecule. **b** Exemplary fluorescent snapshots of nascent proteins localized to surface-immobilized DNA molecules through coupled transcription and translation machines in *E. coli* cell lysate. Co-localization was observed for many DNA molecules and at least $n = 10$ independent replicates.

are spatially coupled. However, direct demonstration of localized protein synthesis requires a single-molecule approach. Whereas transcription of a single DNA and translation of a single mRNA has been studied outside a cell[23–26], observation of newly synthesized proteins tethered to a single DNA has so far remained an experimental challenge.

Here, we use a genetically encoded fluorescent reporter system with improved temporal resolution to observe nascent proteins emerging from single DNA molecules tethered by transient complexes of RNAP, mRNA, and ribosome. We find an accumulation and burst-like production of nascent proteins, a dynamic process seemingly coupled to the DNA topology and gene length. We observe that such localized protein synthesis can facilitate gene regulation and gene expression cascades on the same DNA molecule despite dilute cell-free conditions. By encoding negative and positive feedback on the same DNA molecule, we rationally build a pulsatile genetic circuit driven through the production of only several proteins per hour and gene. Our study presents technical achievements and molecular insights toward single-DNA circuits for nanobiotechnology applications and artificial cell design.

## Results

### Observing nascent proteins on DNA through localized protein synthesis

We set out to explore cell-free protein synthesis on DNA with single-molecule resolution by surface-immobilizing fluorescently labeled DNA molecules at low density in a microfluidic flow channel and imaging with total internal reflection microscopy (TIRFM) (Methods, Supplementary Fig. 1). The native gene expression machines (RNAP and ribosomes) in *Escherichia coli* lysate[27] produced the genetically encoded proteins. To visualize the nascent proteins on DNA through the short-lived RNAP-mRNA-ribosome complex, we used the far-red fluorogenic dye MaP655-Halo, which reacts rapidly with nascent HaloTag (HT) proteins increasing the fluorescence signal by ~1000-fold (Fig. 1b and Supplementary Fig. 1b)[28,29]. The *ht* gene was further fused to C-terminal protein extensions with variable lengths to tune the residence time of the nascent HT proteins on a gene during synthesis. We predicted a reduction of the HT protein signal as the extension is shortened, bound by a cut-off value corresponding to rates of co-translational folding and dye binding (Fig. 2a).

We observed the emergence of persistent HT protein synthesis spots after a few minutes of inflowing the cell lysate and dye (Fig. 2b and Supplementary Fig. 2). The number of HT expression spots was maximal for a 1980 nucleotide (nt) long transcript, reached half of the maximum around 862 nt, and decreased to background levels below a length of 730 nt (Fig. 2c). In contrast, upon replacement of the HT in the 730 nt-long extension construct with Venus, the fastest maturating fluorescent protein[30], we could hardly detect any protein synthesis spots (Supplementary Fig. 3a), suggesting that the fast HT labeling strategy was necessary to report on the short-lived RNAP-mRNA-ribosome complex on DNA.

Over the course of ~1 h, a single DNA produced a few stochastic bursts of HT signal fluctuations, typically lasting from a few (no extension) to tens (longest extension) of minutes each, followed by an extended period of background levels (Fig. 2d). A residence time of nascent proteins for a few minutes is expected with ~1 kb-long genes considering the elongation rates of 10–30 nt per second and 3–10 amino acids per second for transcription and translation, respectively[31]. As HT proteins bleached in around 1.5 min (Supplementary Fig. 1d–h), the longer bursts with stepwise signal jumps suggested the birth and release of multiple HT proteins linked to the same gene simultaneously (Fig. 2d and Supplementary Fig. 3b). We combined the intensity traces from all identified HT synthesis spots into probability distributions (Method, Fig. 2e and Supplementary Fig. 3b). The background levels (i.e., absence of HT proteins on the DNA) produced a background peak in the intensity distribution which reduced with the longer C-terminal extensions. A broad peak/tail appeared and shifted towards higher intensity values for the longer extensions, again indicating that more gene-tethered fluorescent HT proteins could be visualized due to the longer residence times. Because positional fluctuations of the DNA along the TIRF excitation blurred the protein signal during the expression experiment, we could not identify distinct protein sub-peaks in the distributions. Still, the maximal HT signal in the spots was higher than that of single HT proteins immobilized on a surface (Supplementary Fig. 1g), consistent with a few nascent proteins simultaneously linked to the DNA molecule.

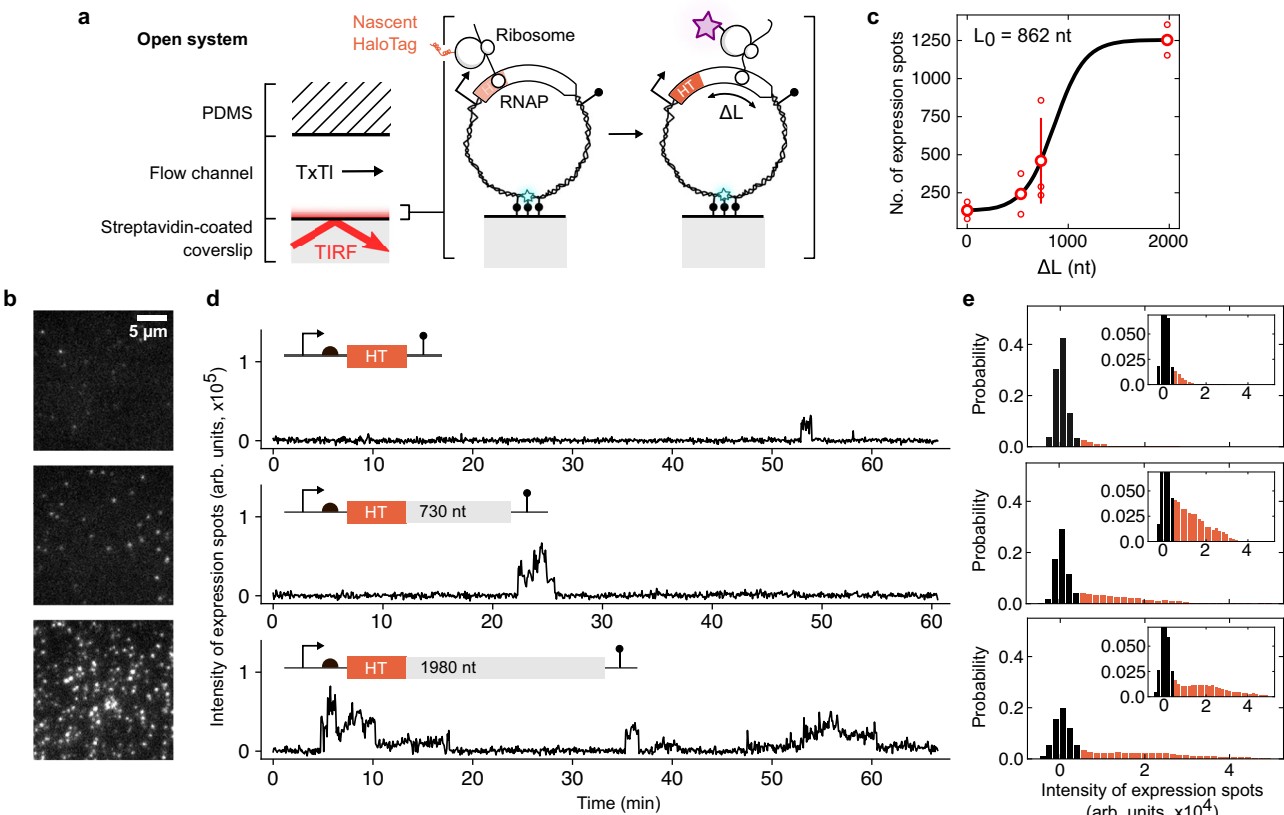

**Fig. 2 | Localized protein synthesis by coupled gene expression machines on single DNA molecules. a** Surface-immobilized DNA encoding the *ht* gene with strong recognition sites for transcription (RNAP) and translation (ribosome) machines from cell lysate (TxTl). The cell lysate and fluorogenic dye were introduced to the DNA under constant flow through a flow channel in an open microfluidic chip. The length of the C-terminal extension fused to the *ht* gene was varied ($\Delta L$), tuning the residence time of HT proteins on DNA. **b** Fluorescent images of the HT signal at the end of expression experiments for the three corresponding extension lengths indicated in d. Scale bar, 5 $\mu$m. **c** The number of expression spots as a function of C-terminal extension length. Experiments were replicated with increasing C-terminal extension length for $n = 2,2,3,2$ (small circles) to obtain average values (large circles). The error bar is given as SD. The gray line is fit to the data with a logistic function and $L_0$ as the midpoint. **d** Examples of protein synthesis traces with the three different nucleotides (nt) extension lengths. **e** Intensity distributions of expression spots over the ensemble corresponding to constructs in **d**. The black bars indicate the background signal. Source data are provided as a Source Data file.

## Transcription-driven protein synthesis on DNA

Having deduced that multiple proteins were present at similar times on the DNA with longer gene extensions raised the question of whether they were formed by a transcription-driven mechanism, in which proteins are synthesized from multiple mRNAs on a single DNA molecule, or translation-driven, with proteins synthesized from multiple ribosomes on single mRNA, i.e., polysomes[32–35]. To discern these two models, we investigated a small fraction of HT synthesis spots (~10%) that gave rise to a slight accumulation of tethered proteins, but only with the longest C-terminal extension (Fig. 3a and Supplementary Fig. 4a, b). We quantified this accumulation with a linear fit to the fluorescent signal and extracted the slope (drift). A similar drift was observed for another construct with a similar extension length (2021 nt) but different sequence, excluding the possibility that protein accumulation was sequence-specific (Fig. 3b). This accumulation of nascent proteins could originate from a transcription-driven mechanism in which RNAPs translocate through the two DNA strands, leading to negative and positive supercoiling waves, which increase the initiation rate and slow down the elongation rate of transcription, respectively[34,36,37].

We reasoned that the accumulation of transcription-induced supercoiling should depend on the number of DNA anchor sites, which constrain the rotation of the DNA and limit the relaxation of super-coiling waves (Fig. 3c)[34,38]. We estimated 1-7 anchor sites per DNA for surface immobilization, based on the DNA sequence at the labeling site

(see Methods for details, Supplementary Fig. 5), and repeated the same experiment with only a single anchor using an alternative labeling protocol (Methods, Fig. 3c). Indeed, we found that the population of accumulating proteins reduced with a single anchor (~3%, Fig. 3b), suggesting that the drift was stemming from topological constraints on protein synthesis due to transcription dynamics. To gain additional support for this transcription-driven protein synthesis across the entire population of HT expression spots, we next supplemented the cell lysate with the antibiotic rifampicin, a selective inhibitor of transcription initiation. We found a strong reduction in the number and intensity of protein synthesis spots (1980 nt) as expected with a reducing number of mRNAs loaded each with a single ribosome (Fig. 3d, e and Supplementary Fig. 3c). If multiple ribosomes synthesized the nascent proteins from a single mRNA, we would have expected a lower number of expression spots, but unaffected high protein intensities per mRNA and DNA molecule.

## Bursts of proteins synthesized on a single DNA molecule

To gain insights into the dynamics of proteins emerging from the DNA through the RNAP-mRNA-ribosome complex, we analyzed the protein fluctuations on single DNA molecules using the ensemble and time averaged autocorrelation function (ACF), $G(\tau) = \langle \delta I(t) \delta I(t + \tau) \rangle / \langle I(t) \rangle^2$, with $\delta I(t) = I(t) - \langle I(t) \rangle$, $I(t)$ the fluorescent intensity at time $t$, and delay time $\tau$. The ACFs exhibited an exponential decay with a correlation time $\tau_C$ and then a drop to minor negative values for $\tau > 10$ min

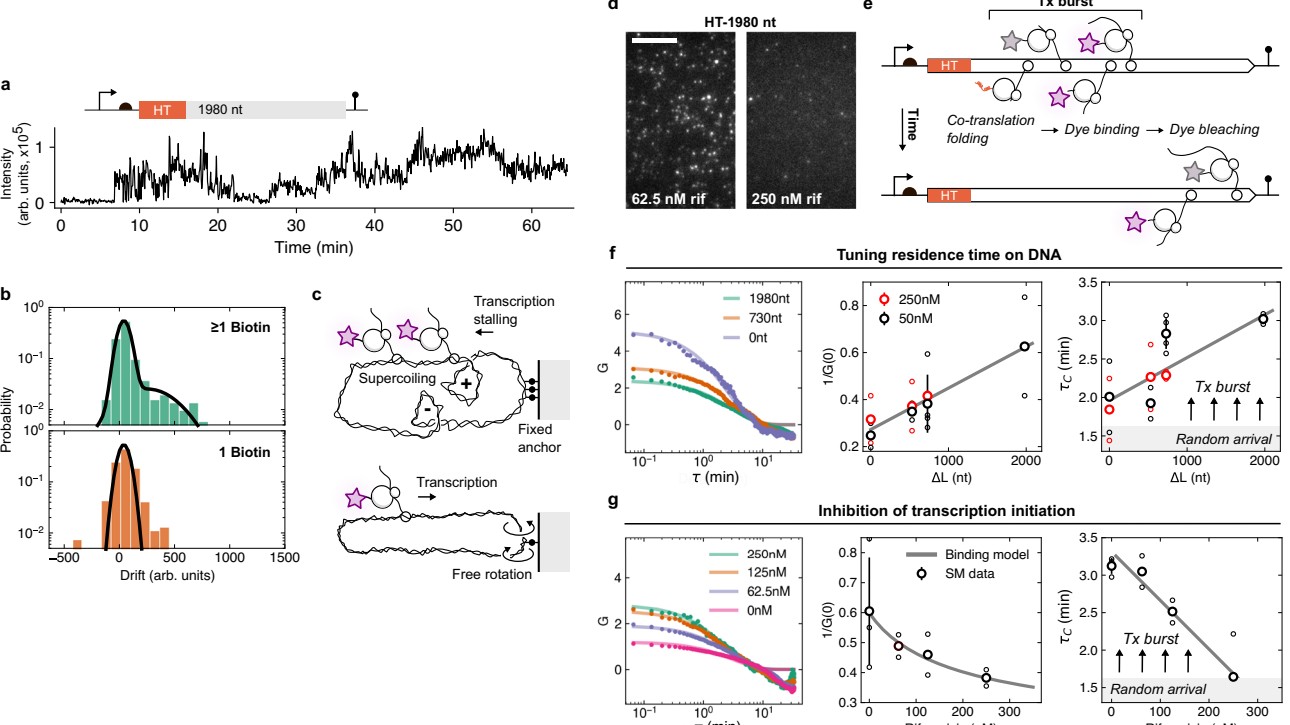

**Fig. 3 | The synthesis rate of nascent proteins by coupled gene expression machines on single DNA molecules. a** Exemplary trace with an accumulation of fluorescent protein signal on a single DNA molecule. **b** Histogram of the accumulation (drift) as computed from linear fits to protein synthesis spots for multiple (top panel) and single (bottom panel) anchor site/s with a 2021 nt-long C-terminal extension. Black lines indicate Gaussian estimates to guide the eye. **c** Schematic depiction of transcription-induced supercoiling. **d** TIRFM images of nascent protein synthesis spots from DNA at two rifampicin (rif) levels with the longest C-terminal extension (1980 nt). Scale bar, 10 µm. **e** Scheme of transcriptional-driven protein synthesis model with nascent proteins (with events of HT folding, dye binding, elongation, and bleaching) produced from DNA via co-transcribing RNAPs in transcriptional (Tx) bursts. **f** Autocorrelation curve (ACF) for HT fused with different C-terminal extensions and fit to mono-exponential decays (solid lines).

The inverse amplitude, $G(0)^{-1}$, and correlation time $\tau_C$ of the ACF for two different concentrations of the fluorogenic dye and four C-terminal extension lengths. A linear trend is shown for both quantities as a gray line. Experiments were replicated with increasing C-terminal extension for $n = 2,2,4,2$ at 50 nM and $n = 2,2,2$ at 250 nM fluorogenic dye (small circles) to obtain the average values (large circles). The error bars are given as SD. **g** ACF for different rifampicin levels with the C-terminal extension of 1,980 nt. The inverse amplitude, $G(0)^{-1}$, followed a binding model between rifampicin and *E. coli* RNAP (solid line). The correlation time $\tau_C$ was extracted from the mono-exponential fit to the ACF. Experiments were replicated with increasing rif concentration for $n = 3,2,2,2$ (small circles) to obtain average values (large circles) with error bars as SD. Source data are provided as a Source Data file.

(Fig. 3f, g), most likely due to finite-sampling effects. The inverse amplitude $G(0)^{-1} = \mu^2/\sigma^2$, and $\tau_C$ increased with DNA extension $\Delta L$, and decreased with rifampicin concentration, where $\mu$ and $\sigma$ are the mean and standard deviation, respectively. The results of the ACF analysis were independent of the number of anchor points per DNA molecule (Supplementary Fig. 4c) and of the HT-dye binding rate (Supplementary Fig. 1c), the latter was verified by increasing the dye level by 5-fold with no apparent effect on the ACF (Fig. 3f).

If signals in the protein time traces were the result of a random birth of a single protein, then $G(0)^{-1}$ and typical timescale $\tau_C$ should be bound by the bleaching time of around 1.5 mins and should not depend on the extension length. We verified this intuitive assumption with numerical simulations and reproduced key features of the ACF with extension lengths and bleaching constraints (see Methods for details, Supplementary Fig. 6). The dependence of $\tau_C$, longer than around 1.5 min, suggested that 1–3 proteins appear in short intervals on a single DNA. The data is consistent with a general scenario of proteins produced in rapid succession, defined as burst-like promoter activity with multiple transcripts and proteins[34,35,39–42]. The model predicted linear scaling between the on-time of a gene and the length, in agreement with our data: $\langle t_{on} \rangle = (\langle \triangle n \rangle - 1)\tau_E + \Delta L/\nu$, given the number of proteins per burst, $\langle \triangle n \rangle$, and the time $\tau_E$ to initiate transcriptional elongation with speed $\nu$ after the E. coli RNAP bound the promoter (right panel in Fig. 3f)[39].

## Cascaded gene expression reaction through localized protein synthesis

The long residence time of nascent proteins in close proximity to their DNA source suggested that we could design DNA nanodevices encoding regulatory proteins that would regulate other genes by a feedback mechanism on the same DNA molecule. We designed a DNA molecule encoding the CI repressor, a key regulator for the life cycle of the lambda bacteriophage[21,43] with its operator binding site at the $P_R$ promoter, inhibiting HT expression upon repressor binding (Fig. 4a). To reduce potential crosstalk of CI repressors synthesized from one DNA molecule and acting on another DNA in the flow channel ($V = 210$ nL), we applied stringent conditions to maintain picomolar concentrations of DNA and proteins by a low DNA surface density (<0.13 DNA µm$^{-2}$) and the renewal of the reaction volume each minute with steady flow (250 nL min$^{-1}$). We surface-immobilized the DNA and after the addition of the cell lysate, strikingly fewer HT synthesis spots appeared than with a DNA lacking the repressor gene (Fig. 4b). The intensity distributions and ACF analysis of the few HT expression spots were similar to those without repressor gene (Supplementary Fig. 7a, b). As additional support for localized regulation on the same DNA molecule, we split the negative cascade into two different DNA molecules, one carrying the *cI* gene, and the other the *ht* gene under CI regulation. Immobilizing both DNA constructs in the same flow channel restored HT expression back to levels without the repressor gene

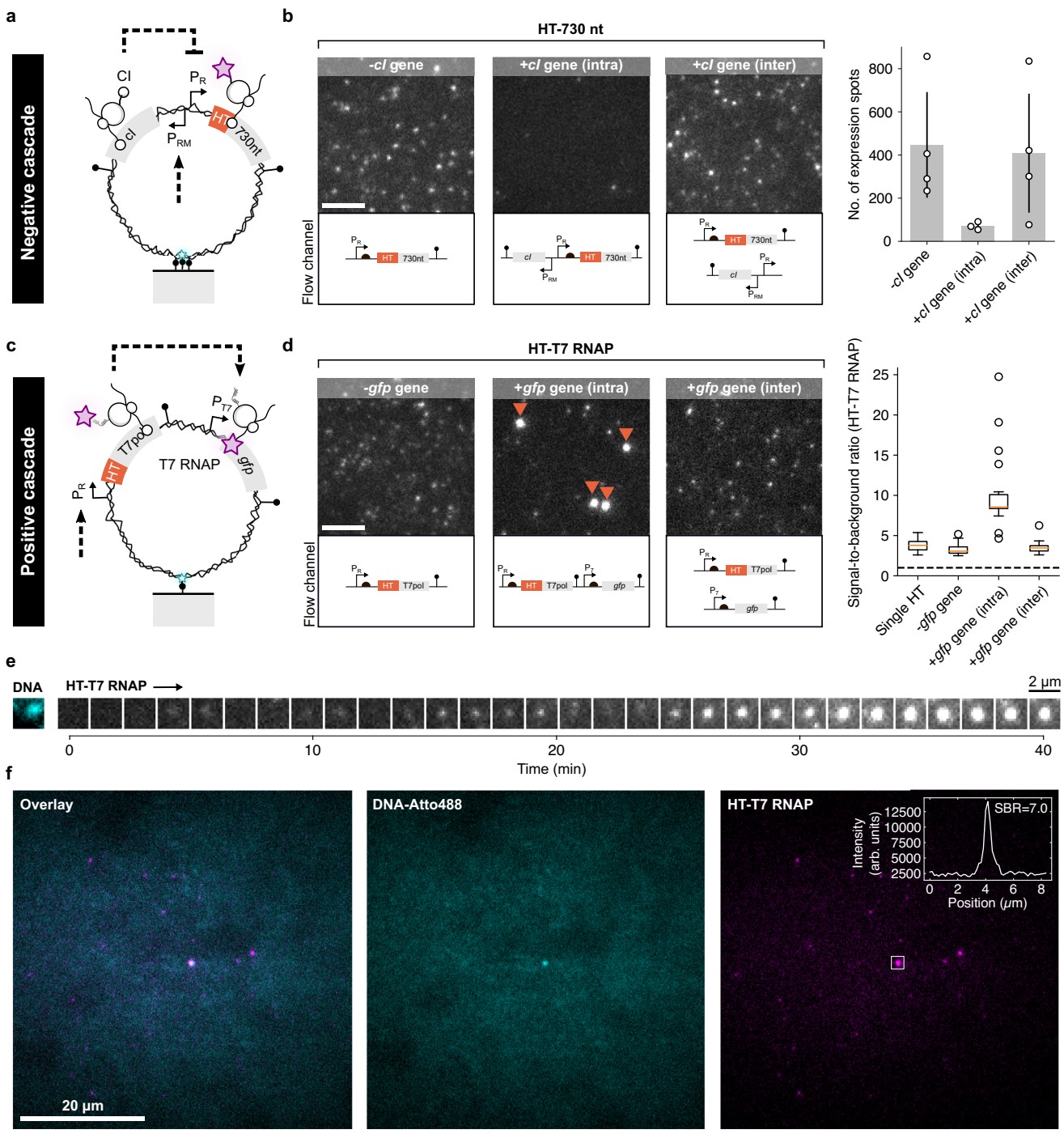

**Fig. 4 | Negative and positive cascaded gene expression reactions on a single DNA molecule. a** Schematic of a DNA molecule encoding the *cl* repressor gene for repression of the HT reporter protein on the same DNA. **b** Fluorescent images of protein synthesis on DNA molecules encoding only the HT protein fused to a 730 nt extension (*n* = 4 independent experiments), with *cl* gene on the same DNA (*n* = 3 independent experiments), and with *cl* gene encoded on a second DNA molecule in the same flow channel (*n* = 4 independent experiments). Scale bar, 5 μm. Number of expression spots with the three different experimental setups. Experiments were replicated (small circles) to obtain average values (gray bars). The error bars are given as SD. **c** Schematic of a DNA molecule producing a T7 RNAP (T7pol) to drive a downstream *gfp* gene under the control of a T7 promoter (P$_{T7}$) on the same DNA. **d** Exemplary fluorescent images of protein synthesis on DNA encoding the gene for the T7 RNAP without *gfp* gene (*n* = 5 independent experiments), with encoded T7 RNAP and *gfp* gene (*n* = 4 independent experiments), and the genes for T7 RNAP

and GFP split between two DNA molecules (*n* = 3 independent experiments) immobilized in the same flow channel. Scale bar, 5 μm. DNA molecules with protein build-up are indicated by red arrow heads. The signal-to-background ratio (SBR) of single dye-bound HT proteins surface-immobilized with antibodies and manually extracted gene expression spots with the three different experimental setups (from left to right, *n* = 21,26,18,13 expression spots over 2 independent experiments). Boxplots indicate median (red line), first and third quartile (black box), and 1.5 times within the inter-quartile range (whiskers) with individual data points shown as circles. **e** Fluorescence image of a DNA molecule and time series of HT-T7 RNAP accumulating on the DNA. Similar traces were observed for many DNA molecules and *n* = 4 independent experiments. **f** Fluorescence images of a very dilute single DNA molecule co-localized to HT-T7 RNAP signal with high SBR. Data was replicated with two independent expression experiments. Source data are provided as a Source Data file.

(Fig. 4b), suggesting that CI repression could not occur at the dilute regime unless the two genes are encoded on the same DNA molecule. Finally, to directly demonstrate co-expressional localization of repressor binding to DNA, we fused the CI gene to the HT protein on single DNA molecules encoding the gene and its target site and observed CI-HT signal co-localized with DNA (Supplementary Fig. 7c).

We next designed a positive gene expression cascade in which the RNAP of the T7 bacteriophage, fused to HT for visualization (HT-T7 RNAP), could activate a downstream gene on the same DNA (Fig. 4c and Supplementary Fig. 8a, b). We first validated the design of the circuit by adding *gfp* as a downstream reporter gene and measuring the dynamics in bulk solution, either combined on the same DNA or split between two DNA molecules (Supplementary Fig. 8b). The HT-T7 RNAP signal was closely followed by the GFP signal with both genes encoded on the same DNA molecule. When the genes were split, the GFP signal increased only slightly and delayed for hours after the appearance of the strong HT-T7 RNAP signal, suggesting that the proximity of nascent T7 RNAP synthesis on the same DNA encoding GFP greatly facilitated GFP synthesis in bulk reactions.

To measure the effect in the single-molecule regime with constant removal of nascent proteins, we immobilized the DNA constructs on the surface using a single anchor point to reduce the topological constraints anticipated due to the expression of the long HT-T7 RNAP gene (2649 nt). A construct coding only for HT-T7 RNAP with no downstream gene resulted in expression spots of nascent HT-T7 RNAP tethered to the DNA through the RNAP-mRNA-ribosome complex (Fig. 4d and Supplementary Fig. 8a). The addition of a *gfp* gene under the regulation of a T7 promoter to the same DNA molecule led to a strong increase in HT-T7 RNAP fluorescence signal on individual DNA molecules. In comparison, no increase was observed when the two genes were split between two DNA molecules immobilized in the same flow channel (Fig. 4d and Supplementary Fig. 8a). The fluorescence signal produced from HT-T7 RNAP with downstream *gfp* gene reached high steady-state HT fluorescence values after few tens of minutes (Fig. 4e and Supplementary Fig. 8a), with ~3-fold higher signal-to-background ratio (SBR) than all other cases (Fig. 4d) suggesting that HT-T7 RNAP was engaged in transcription of the downstream gene. We also found such high SBRs in very dilute conditions with only a single DNA molecule in a large ~2500 μm² region, that is, about a 1000-fold lower concentration of DNA and nascent proteins than previously (Fig. 4f). These observations suggested that the nascent T7 RNAP could interact with the T7 promoter to start transcribing the *gfp* gene on the same DNA, during or shortly after its own synthesis.

### A genetic feedback circuit on a single DNA molecule

Finally, we constructed an entire gene circuit with activator and repressor genes in feedback on the same DNA. We used the positive cascade construct (Fig. 4c) as a basis, fused the *gfp* gene to a gene encoding the strong synthetic dCro repressor[44], and replaced the promoter regulating HT-T7 RNAP expression with a strong de novo designed synthetic promoter[45], that could be repressed by dCro (see Methods for the design details, Fig. 5a and Supplementary Fig. 9a). We validated the construct in bulk by comparing to a construct with mCro repressor replacing dCro (Supplementary Fig. 9b). The mCro is the monomeric form of the Cro repressor and is therefore much weaker than the synthetically fused dCro dimer. mCro is also weaker than the CI repressor used in the negative cascade (Fig. 4a, b and Supplementary Fig. 9c). In the bulk reaction the HT-T7 RNAP signal increased during the first tens of minutes and plateaued with an increase of the dCro-GFP signal, suggesting fast repression of HT-T7 RNAP by dCro-GFP. In contrast, the mCro-GFP construct continued to accumulate HT-T7 RNAP for the rest of the experiment without apparent repression with an increase in the mCro-GFP signal (Supplementary Fig. 9c).

To directly demonstrate the gene circuit driven by localized protein synthesis, we next measured the constructs at the single-molecule level where dissociated nascent proteins are quickly removed from DNA (Fig. 5b). Individual HT-T7 RNAP synthesis traces from the genetic circuits with either dCro (Supplementary Movie 1) or mCro (Supplementary Movie 2) showed fluctuations on many time scales. We computed the power spectrum in the frequency domain $f$ and observed a power-law decay reminiscent of $1/f$ noise (Fig. 5c). The spectrum was identical for the mCro and dCro constructs, except at the lowest frequency $f_{min}$ where mCro has a slightly higher amplitude. To identify pulsation, we plotted the maximal Fourier amplitude ($\mathcal{F}_{max}$) and linear drift (Fig. 5d–f). We expected high $\mathcal{F}_{max}$ and low drift values for these traces as they returned to background levels after a pulse of HT-T7 RNAP production (see Supplementary Fig. 9d for details). We found a bimodal distribution in drift values for the construct with mCro repressor. We identified the high-drift peak corresponding to HT-T7 RNAP accumulation on DNA, consistent with lack of repression (class 1 and 2, Fig. 5d). The high-drift peak was absent in the drift distribution of the dCro traces, indeed giving rise to traces with high $\mathcal{F}_{max}$ and low drift, indicative of repression (class 3, Fig. 5d, f). A pulsing factor, defined as the ratio between drift and $\mathcal{F}_{max}$ was significantly different between the mCro and dCro constructs (Fig. 5e). About 20% of the dCro traces, and only about 3% among the mCro traces, showed pulsatile characteristics with a rise and decay of HT-T7 RNAP signal in ~40 min, suggesting the delayed repression of HT-T7 RNAP by dCro-GFP even at the single-molecule level with only several proteins in positive and negative feedback.

## Discussion

We present a methodology to observe gene expression from single DNA molecules outside a living cell, combining the HT protein and fluorogenic MaP655-Halo dye with single-molecule TIRFM. The ability to detect nascent proteins as they emerge from single DNA provided molecular insights on the coupling of *E. coli* transcription and translation machines. We could reconstitute gene regulation on single DNA molecules by a co-expressional localization mechanism that has not been demonstrated previously. With the synthesis of only several proteins per hour and gene at very dilute cell-free conditions, the complex genetic circuits that we demonstrated here required coupled transcription and translation, co-translational protein folding, and protein binding to the DNA target site, all happening on the same DNA molecule in an autonomous fashion. Such dissipative nanodevices are readily produced with standard molecular biology techniques and executed with only cell lysate and a single DNA molecule (Supplementary Data 1).

Commonly, engineered gene circuits are driven by hundreds or thousands of protein copies in living and artificial cells[12,13,21,46]. The large number buffers fluctuations and often improves their function through a fast search of the regulatory proteins for their target sites on DNA. Despite the much lower number of nascent proteins, our cell-free genetic circuits performed programmable pulsatile behavior on a large fraction of DNA molecules. Variability between DNA molecules mostly came from DNA molecules with apparent low expression activities, which could stem from issues with surface immobilization, e.g. DNA surface-adsorption and therefore reduced accessibility for the gene expression machines.

Our work may inspire the construction of dissipative autonomous nanodevices with more functions at the nanoscale. The DNA molecules could be embedded together with DNA origami platforms[9], programmed with the rich toolbox of cell-free DNA replication, editing, and recombination[47–49], and used as genetic foundation for the construction of artificial cells with optimized function and resource load, whether the DNA is dispersed in bulk solution[10,11,50], encapsulated inside membrane vesicles[51], or embedded in two-dimensional solid compartments[12,13].

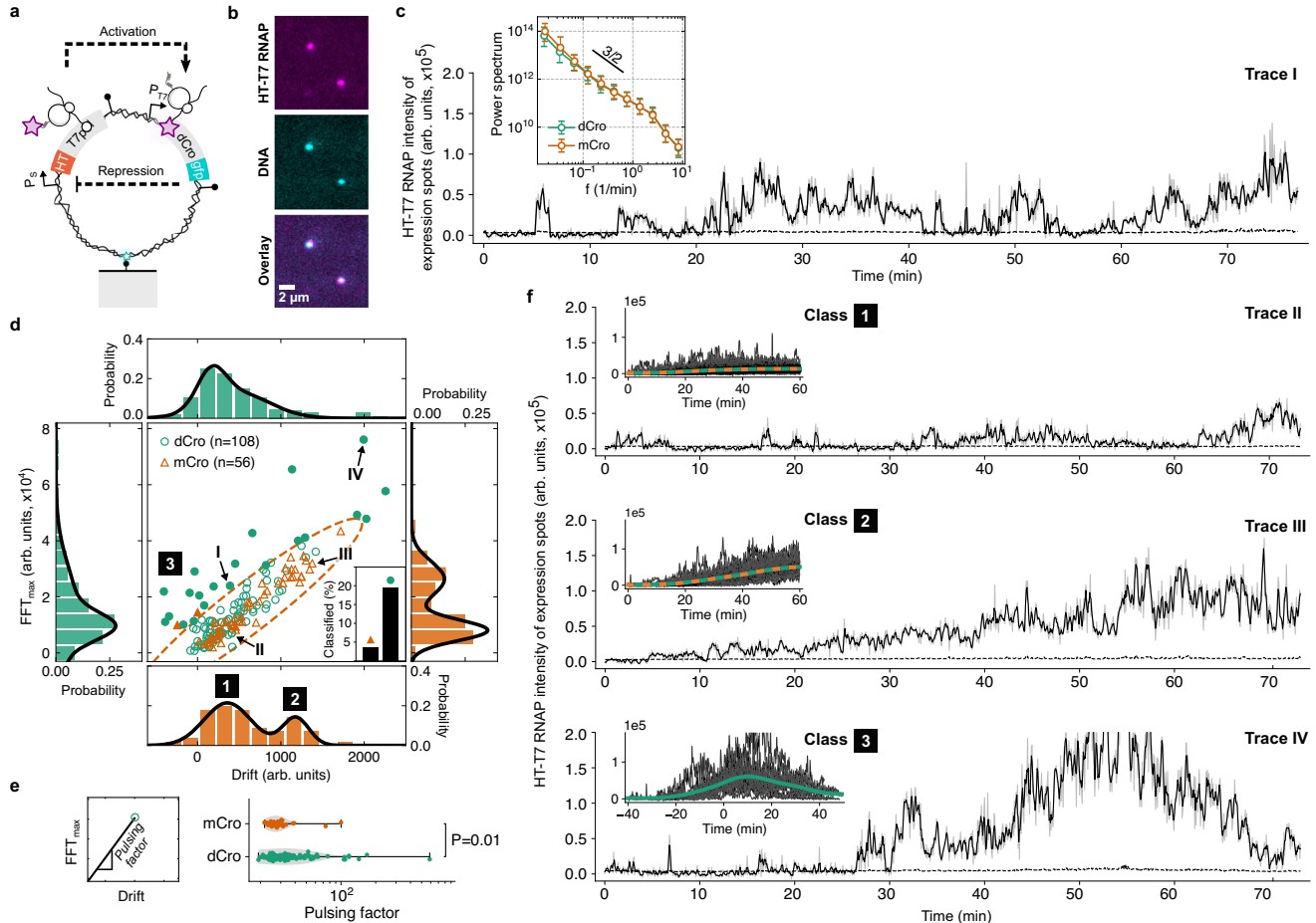

**Fig. 5 | Genetic circuit with positive and negative feedback on a single DNA molecule. a** Schematic of a DNA molecule encoding the gene for T7 RNAP producing a downstream strong single-chain dimeric Cro (dCro) repressor to repress HT-T7 RNAP synthesis. **b** Two-color fluorescent images of HT-T7 RNAP signal co-localized to two surface-immobilized DNA molecules. Scale bar, 2 μm. **c** Exemplary trace of a single DNA molecule (shown in panel b) encoding the gene circuit in panel a with dCro and highlighted in panel **d** (Trace I). The black line is a moving average over the raw data in light gray. A second construct with a weak monomeric Cro (mCro) repressor, replacing dCro, was probed as a proxy for no/weak negative feedback. The HT-T7 RNAP expression signals were transformed into frequency space using the Fast-Fourier-Transformation (FFT). The ensemble-averaged power spectrum (the black line indicates $f^{-3/2}$ decay) is given as median and error bars representing 32th and 68th percentiles from a total number of DNA molecules ($n = 108$ for dCro and $n = 56$ for mCro) examined over $n = 5$ (dCro) and $n = 4$ (mCro) independent experiments. **d** Scatter plot showing the maximal FFT amplitude ($\mathscr{F}_{max}$) against signal drift computed as a linear trend in the HT-T7 RNAP expression

traces. The dashed ellipse estimates the reference distribution without negative feedback from the construct with mCro. Traces outside the ellipse were classified as pulses (filled markers). The inset shows the fraction of HT-T7 RNAP expression traces classified as pulses for the two constructs. Projected histograms are overlaid with bimodal Gaussian fits to guide the eye. The same number of DNA molecules were used as in panel **c. e** Pulsing factor is defined as the ratio between $\mathscr{F}_{max}$ and signal accumulation (drift). The pulsing factor distributions for the two constructs from panel d, excluding the traces with negative HT-T7 RNAP signal accumulation. The violin plot shows the Gaussian kernel density estimate as gray area and error bars indicate minimum and maximum values. The $p$-value is computed by two-sided Welch's $t$-test. The same number of DNA molecules were used as in panel **c. f** Exemplary HT-T7 RNAP synthesis traces for the three highlighted traces in panel d (Trace II–IV). The insets show individual traces overlaid with the average signal found from population classes in panel d (Class 1–3). Individual pulses were aligned in time with the phase shift computed at $\mathscr{F}_{max}$. Source data are provided as a Source Data file.

## Methods

### Preparation of Escherichia coli lysate

The cell lysate was prepared from *Escherichia coli* (BL21 Rosetta 2) strains[27]. Briefly, bacteria were grown in 2xYT supplemented with phosphates. With an OD600 of 1.5-2, the bacteria were lysed using a pressure cell. After the first centrifugation step ($12,000 \times g$ for 10 min), the supernatant was incubated at 37 °C for 80 min to digest residual nucleic acids using endogenous exonucleases. After a second centrifugation step ($12,000 \times g$ for 10 min), the supernatant was dialyzed for 3 h at 4 °C. After a final spin-down of the dialyzed crude cell lysate ($12,000 \times g$ for 10 min), the supernatant was aspirated, aliquoted (29 μL), and stored at −80 °C. Before use in cell-free gene expression experiments, the cell lysate was thawed on ice, supplemented with auxiliary solutions (10 mM Mg-glutamate, 80 mM K-glutamate, 4% PEG8000, 10.8 mg mL$^{-1}$ Maltodextrin, amino acids, energy regeneration, nuclease

inhibitor GamS), filled to 78.3 μL with water, and gently mixed with a pipette. Finally, a volume ratio of 9 parts cell lysate and 1 part water containing reagents, e.g. DNA, fluorogenic dye, and antibiotics, were mixed.

### Construction and preparation of DNA molecules

The primers were ordered at IDT. Genes were amplified with KAPA HiFi HotStart ReadyMix (Kapa Biosystems, Roche). The DNA molecules were derived from P70a-UTR1-deGFP[27], cloned with Gibson assembly, and sequenced with the Weizmann in-house sequencing service and Nanopore sequencing provided by Plasmidsaurus (SNPsaurus LLC). We introduced sites for the methyltransferase (MT) and nicking enzyme between the ampicillin (Amp) resistance gene and ColE1 origin of replication (Ori) of the pBEST vector. We replaced the *degfp* gene with the *ht9* gene[28] containing silent mutations in order to remove MT

and nicking sites on the open-reading frame. The C-terminal fusion tags were cloned as complete or truncated genes from T7 *gp3* (*L* = 531 nt), *gp48* (truncated version, *L* = 730 nt), *gp6* (*L* = 1980 nt), and *gp7* (truncated version, *L* = 2021 nt)[52], and fused to the *ht* gene with a short peptide linker (KRAPGTS).

The *cI* gene and regulatory parts were lifted from the genome of the lambda bacteriophage as previously described[21] and cloned into the pBEST plasmid encoding *ht-gp48*. To directly observe the dynamics of CI on DNA, we fused HT (linked by the peptide GGGGSGGGGS) to the C-terminus of CI for two reasons: (1) to reduce interference with the N-terminal DNA binding domain[21] and (2) to maintain the leaderless structure of the *cI* mRNA and corresponding translation rate for both CI constructs with and without fusion. The leaderless mRNA starts directly at the AUG initiation codon without an extra ribosomal binding site. Notably, CI is assumed to predominantly bind after dimerization in solution to its target site on DNA[43]. In the very dilute conditions of our cell-free expression experiments, dimerization seems unlikely. Transient gene regulation may however come from CI monomers briefly binding to the operators during or shortly after synthesis on the same DNA[53].

To remove the requirements for dimerization and strengthen the repression for the full gene circuit, we implemented a single-chain dimeric Cro (dCro) repressor gene (the monomers were linked by a peptide AAGGGTGGGS). The gene was ordered as gBlock (IDT) and fused to the C-terminus of GFP (linked by the peptide RRAKKEA)[12] located on the construct encoding the positive cascade with HT-T7 RNAP. The construct with monomeric Cro (mCro) was prepared by removing the first N-terminal Cro domain. The synthetic promoter with two strong binding sites for the Cro repressor[54] was designed with "Promoter Calculator"[45] and the sequence TCTATCACCGCGGGTGAT AAANNNNNNNNNNTATCCCTTGCGGTGATANNNNNNNNN

towards maximizing transcription rates. First, we designed a total of 3 different synthetic promoters (also with weaker binding sites) and verified their activity with the constitutive expression of GFP in solution (Supplementary Fig. 9a and Supplementary Table 1). We chose the strongest promoter ($P_{S.VI}$) and replaced the native $P_R$ promoter with the synthetic promoter to construct the full pulsatile gene circuit (see Supplementary Data 1 for their DNA sequences).

For all experiments, the DNA was first propagated in DH5α and then purified from 10−15 mL of KL740[27] in LB medium grown at 30 °C with Ampicillin. After purification with a spin column (Promega), 5 μg of DNA were nicked with 5 units of Nb.BbvCI (NEB) in CutSmart buffer (NEB) for 1.5 h at 37 °C. The volume was 20 μL. After inactivation of the nicking enzyme at 80 °C for 20 min, we added all four nucleotides at a final concentration of 1 μM (Biotin-7-dATP, Jena Biosciences, and dC-Atto647N or dU-Atto488, Jena Biosciences with respective pair of unmodified nucleotides, Thermo Scientific): [Biotin-7-dATP, dT, dC-Atto647N, dG] and [Biotin-7-dATP, dU-Atto488, dC, dG]. Next, we added 7.5 units of *E. coli* DNA Polymerase I (NEB) and incubated the mix for 3 h at 20 °C. Finally, the modified DNA was purified with ethanol precipitation and kept at −20 °C for long-term storage. With our labeling protocol, multiple biotins are introduced in front of the gene, leaving a single nick at the promoter site (Supplementary Fig. 5a). This asymmetry would constrain positive supercoiling and stall transcription elongation as discussed in the main text. We performed cell-free gene expression from nicked and un-nicked DNA without the nuclease inhibitor GamS and observed identical expression dynamics (Supplementary Fig. 5b), suggesting that the DNA stability is not influenced by the nick.

For DNA used to study the supercoiling effects through the number of biotin immobilization sites (*ht9* gene fused to 2021 nt-long truncated *gp7* gene), positive cascades, and full gene circuit, we used an MT enzyme in combination with biotinylated cofactor AdoYnBiotin. The MT enzyme transfers a single biotinylated methyl group from the cofactor onto a specific DNA site (ATCGAT)[55]. 5 μg of DNA was treated with M.BseCI and 80 μM biotinylating cofactor AdoYnBiotin[56] in the MT reaction buffer (10 mM Tris-HCl, 10 mM EDTA, 5 mM β-Mercaptoethanol, pH 7). The solution was incubated in a total volume of 50 μL for 1 at 55 °C, and then the column (Promega) was purified. We kept the nicking configuration as previous DNA molecules by further nicking and nick-translating DNA with a nucleotide mix as described above, substituting the Biotin-7-dATP with dA (Sigma-Aldrich). Anticipating a noisier response from the full genetic circuits, we also improved our labeling methodology to incorporate more fluorescent nucleotides, readily identifying all DNA molecules on the surface. We nick-translated the DNA with a final nucleotide concentration of 25 μM dA, dC, dG, and 15 μM dT, and 10 μM dU-Atto488. The DNA was purified with ethanol precipitation and kept at −20 °C for long-term storage.

### Design, fabrication, and preparation of microfluidic chips

We designed the microfluidic chip layout with AutoCAD 2021 (Autodesk) and exposed 5" chrome masks (Nanofilm) with μPG 101 (Heidelberg). The chrome mask was developed according to the manufacturer's protocol. Next, a 4" silicon wafer (0.525 mm thickness, <100>, p-type, University Wafers) was cleaned with acetone, isopropanol, and water. The wafer was again cleaned with the plasma system (250W, $O_2$ at 40 sccm, 150 mTorr, 120 s; March AP-300, Nordson) and pre-treated with Hexamethyldisilazane (HMDS, Transene Company) for 30 s to promote the adhesion of the SU-8 photoresist on the wafer. Residual HMDS was removed from the wafer by spinning at 3000 rpm for 60 s. The SU-8 3050 resist was spun on the silicon wafer: (1) 7.5 s at 500 rpm, (2) 30 s at 3000 rpm. The resist was exposed with the mask and mask aligner (Karl Suss Ma6/BA6), baked, and developed according to the manufacturer's protocol. After verifying the structures, the resist was hard baked at 150 °C, and slowly cooled down to room temperature. The final dimensions were measured with a profilometer (Dektak, Bruker) and optical microscopy.

The silicon wafer with fabricated SU-8 features was placed in a petri dish and covered with ~40 mL of polydimethylsiloxane (10:1 PDMS: Curing agent, Sylgard 184, Dow Corning). The dish was placed for ~1 h under vacuum to remove bubbles and baked at 75–80 °C for ~4 h. The PDMS was cut into pieces using a scalpel and peeled off the wafer. The holes for the inlet and outlet were punched (0.75 mm diameter, Welltech Labs) on a cutting mat. The microfluidic chips were cleaned with isopropanol and blow-dried. The coverslips (#1.5, Marienfeld) were cleaned by boiling in (1) 96% ethanol for 10 min and 70 °C warm and (2) 3:1:1 $H_2O$: $NH_3(25\%)$: $H_2O_2$. The coverslips were blow-dried and stored in a dust-free box. Just before bonding PDMS to the coverslip, we cleaned coverslips a final time with the plasma system (250 W, $O_2$ at 40 sccm, 150 mTorr, 120 s). The PDMS slabs and coverslips were surface activated (100 W, $O_2$ at 49 sccm, 180 mTorr, 20 s) and brought in contact for permanent bonding. The assembled microfluidic chips were baked at 75–80 °C for ~4 h.

For surface passivation, the microfluidic chips were incubated with 0.15% Dichlorodimethylsilane (DDS, Sigma-Aldrich)[57] in 1.5 mL anhydrous acetonitrile (Sigma-Aldrich) for ~30 min. After the treatment, the microfluidic chips were successively washed with 1.5 mL of acetonitrile, 1.5 mL equimolar mix of acetonitrile and water, and 1.5 mL of water. The microfluidic chips were kept wet in a humid chamber until use (typically for 0–2 days).

### Single-molecule microscope

Fluorescent images were acquired using Micro-Manager 2.0.0 and a custom-built single-molecule TIRF microscope as previously described[21]. The lasers with 488 nm (100 mW, OBIS, Coherent) and 647 nm (120 mW, OBIS, Coherent) excitation were combined (DMSP605 and 5xBB1-E02, Thorlabs) via an objective mounted on an XYZ stage (MBT612D/M, Thorlabs) into single-mode fiber (P5-460B-PCAPC−1, Thorlabs). The fiber was coupled into a mirror collimator (RC08FC-P01, Thorlabs) to expand the laser diameter to 8 mm, guided

through an achromatic lens ($f$ = 150 mm, AC254–150-A-ML, Thorlabs), redirected by a mirror and filter cube (TRF59906, Chroma), and focused onto the back focal plane of the TIRF objective (60x, 1.49 NA, Nikon). The emission was focused with a tube lens (TTL200, Thorlabs) onto an EM-CCD (iXon Ultra 888, Andor Technology, Belfast, UK). The two lasers were controlled with an Arduino microcontroller (Arduino control software v1.2) and synchronized with the output trigger signal of the EM-CCD camera. The sample was translated in XY (Märzhäuser Wetzlar), and the TIRF objective was mounted with an in-house build Delrin adapter for thermal insulation on a piezo stage to focus in Z (400 μm Fast PIFOC, PI). The objective was further enclosed with resistive heating foil (HT10K, Thorlabs) to set the temperature at 34 °C using a PID controller (TE-48-20, TE Technology). We designed a custom humidity chamber and constantly flushed it with humid $N_2$.

### Preparation of the fluorogenic dye MaP655-Halo

The fluorogenic dye MaP655-Halo was synthesized as described previously (where the fluorogenic dye was initially named "Probe 29")[29], stored at −20 °C in anhydrous dimethyl sulfoxide (DMSO, Sigma-Aldrich), and diluted in water (+/− DNA) to a 10-fold stock solution just before the expression experiments were started. Briefly, MaP655-Halo was synthesized upon protection of the 6-carboxy group of 6-carboxy silicon rhodamine using allyl bromide in the presence of $Et_3N$ and $K_2CO_3$. Subsequently, the acyl chloride was formed by means of $POCl_3$ followed by nucleophilic acyl substitution with 3,5-difluorobenzenesulfonamide. Deprotection of the allyl ester using $Pd(PPh_3)_4$ and 1,3-dimethylbarbituric acid as well as coupling of the chloroalkane ligand to the 6-carboxy group by means of PyBOP and DIPEA yielded MaP655-Halo.

### Performing gene expression experiments in solution

For gene expression in solution, 1 μL of a 10-fold stock solution (containing 10 nM of each DNA) in water was mixed with 9 μL of cell lysate and expressed at 32 °C in a 96-well plate (Costar) tightly sealed with a plastic lid (Cat. No. 3080, Costar). For DNA encoding the *ht* gene, the DNA stock solution was supplemented with 10-fold fluorogenic dye (typically 200 μM). The fluorescent signals were recorded every 2 min using a plate reader (ClarioStar Plus, BMG Labtech).

### Performing single-molecule experiments on the surface

Before use, a DDS biochip was first treated for ~15 min with 0.2 mg mL⁻¹ biotin-BSA (Sigma-Aldrich) in PBS and, secondly, for ~15 min with 0.2% Tween-20 in PBS. Meanwhile, 3 nM of fluorescently labeled and biotinylated DNA (or 2x1.5 nM with experiments for the intermolecular cases) were bound to 5 nM Streptavidin in PBS and with 1:1800 Tetraspeck beads (0.1 μm, ThermoFisher). After ~10 min incubation time, the mixture was flushed on the DDS chip and incubated for ~15 min. Non-specifically adsorbed Tetraspeck beads were later used as fiducial markers to correct the lateral drift during the expression experiments. The residual DNA and beads were removed from the flow channel by flushing with fresh PBS. A 10–20 μL mixture of cell lysate, fluorogenic dye (typically at a final concentration of 50 nM if not stated otherwise), and optional rifampicin (stock solution was stored in anhydrous DMSO at −20 °C, Sigma-Aldrich) was prepared and pulled from a 10 μL pipette tip attached to the inlet port through the microfluidic chip with a glass syringe (Gastight, Hamilton) and syringe pump (PHD Ultra, Harvard Apparatus) at a constant flow rate of 0.25 μL min⁻¹. Fluorescent images were acquired with an excitation power of 15 W cm⁻² (488 nm) and 15 W cm⁻² (647 nm) at a frame rate of 0.5 Hz (in alternating mode between the two excitation wavelengths; time delay between frames of the same excitation wavelength was therefore 4 s), 100 ms exposure time, and 750 gain.

### Pull-down of individual HT proteins produced in cell lysate

We mixed the biotinylated high-affinity (3F10) anti-HA antibodies (50 μg mL⁻¹, ~500 nM, Sigma-Aldrich) at a 2:1 ratio with Streptavidin (Sigma-Aldrich) in PBS and incubated the solution on ice for ~30 min. Antibodies were bound to the DDS surface for ~20 min. The chip was thoroughly flushed with PBS to remove unbound antibodies and used on the same day. The DNA encoding the HA-HT construct was expressed in 10 μL cell lysate at 30 °C with 50 nM MaP655-Halo for 1 h. After this time, the plateau in the cell-free expression curves (Supplementary Fig. 1c) suggested that all fluorogenic dyes reacted with HA-HT proteins. The lysate was diluted by 1000-fold in PBS, flushed on the anti-HA antibody-coated coverslip, incubated for a few minutes, and thoroughly washed with fresh PBS. The individual surface-immobilized HA-HT proteins bound to fluorescent MaP655-Halo were imaged with 10, 20, and 40 W cm⁻² at 647 nm, a frame rate of 10 Hz, and an exposure time of 100 ms.

### Single-molecule image processing for experiments on the surface

The automated processing was implemented with Python v3.7 and the following packages: Matplotlib v3.3, NumPy v1.20.3, Scipy v1.3, scikit-image v0.15, numba v0.53.1. First, single-molecule images were corrected for lateral drift by detecting and fitting a 2-d Gaussian function to the non-specifically adsorbed Tetraspeck beads (0.1 μm, Thermo-Fisher). Usually, we imaged regions with ~3 beads. After correction, the protein and DNA spots were identified in each frame with a custom adaptation of Picasso[58], localized by the radial symmetry method[59], and linked with trackpy v0.5.0[60] (Supplementary Fig. 2). For gene expression experiments, we removed spurious protein appearances that occurred for less than 5 frames or were outside of clusters as defined by the density-based spatial clustering algorithm (scikit-learn v0.24.2, dbscan). The protein clusters, i.e. protein synthesis spots, were labeled as co-localized with DNA when a DNA molecule spot was closer than 7 pixels (~1 μm) (Supplementary Fig. 2b). The intensity traces were measured for each protein synthesis spot and fluorescent spot (for the DNA bleaching traces) from a circular region of diameter $d$ with 3 pixels (~0.5 μm) subtracted by the sigma-clipped (low=2, high=2) background ring ranging from $d$ to $2d$.

As for the downstream analysis of protein synthesis spots, we counted 131 HT spots for the shortest C-terminal extension length (Fig. 2c) and attributed them to non-specifically surface adsorbed free and HT-bound dye. Assuming this unspecific adsorption does not change with C-terminal extensions, these should be subtracted from the 1123 spots that appeared for the longest extension (Fig. 2c), suggesting that ~88% of the spots were nascent proteins tethered to coding genes. About 30% of all HT expression spots could be directly co-localized to DNA molecules (Supplementary Fig. 2b), the rest ought to come from unidentified DNA (~58%). Except for the full circuit (where we improved the DNA labeling to increase the number of fluorescent nucleotides per DNA molecule), we therefore included all protein synthesis spots for the downstream analysis. We reasoned that the considerable fraction of unidentified DNA molecules on the surface could come from two potential issues: (1) Immobilization of non-fluorescent DNA molecules due to no/low dye incorporation and the bleaching of the fluorescent nucleotides during the preparation steps. (2) The auto-fluorescent cell lysate masked low-signal DNA molecules. For the positive cascade, intensities of expression spots were manually extracted with line profiles using the Fiji distribution of ImageJ 1.51p.

### Interpretation of the autocorrelation function

We computed the autocorrelation function (ACF) from each expression spot and analyzed the median ACF. The median ACF gave a quantitative interpretation of the ensemble of fluctuations in the protein synthesis signal. We computed the ACF according to

$$G(\tau) = \frac{\langle \delta I(t) \delta I(t+\tau) \rangle}{\langle I(t) \rangle^2} \tag{1}$$

with $\delta I(t) = I(t) - \langle I(t) \rangle$, $I(t)$ the fluorescent intensity at time $t$, and the correlation time $\tau$. At $\tau = 0$, $G(0)^{-1}$ becomes

$$G(0)^{-1} = \frac{\mu^2}{\sigma^2} \qquad (2)$$

with $\sigma^2$ and $\mu$ as the variance and average of the intensity distribution, respectively. In the ideal case of fluctuations without correlated protein production and fluorophore bleaching[61]

$$G(\tau) = \frac{T - \tau}{k_{tx}T^2} H[T - \tau] \qquad (3)$$

and

$$G(0)^{-1} = k_{tx}T \qquad (4)$$

with $H$ as Heaviside function, $k_{tx}$ as transcription initiation rate and $T$ as the residence time of a protein on its gene. We tested this ideal model with our experimental limitations using Monte-Carlo simulations (Supplementary Fig. 6). In the following paragraph, we describe the model for nascent fluorescent HT proteins emerging from a DNA molecule with the steps of transcription-translation initiation, elongation, co-translational peptide folding, dye binding and bleaching, and termination (Supplementary Fig. 6b):

We considered the RNAP-mRNA-ribosome-protein complex as directly coupled (simultaneous start of both machines), initiating coupled transcription-translation events with exponentially distributed waiting times to start moving on the gene with length $\triangle L$ and an elongation rate $v$. We assumed the complex's total residence time $T(=\triangle L/v)$ on the gene to be normally distributed ($X \sim \mathcal{N}(T, (0.1T)^2)$) considering the central limit theorem with the many intermediate steps during coupled transcription and translation. While moving on the gene, the time a nascent HT protein requires to start fluorescing after the transcription-translation complex passes the end of the *ht* gene and starts decoding the C-terminal extension before termination is given by the time of co-translational folding and dye binding. The binding time between HT protein and dye is estimated at around 2 s considering a binding rate of $k \sim 10^7$ M$^{-1}$s$^{-1}$[28], with 50 nM MaP655-Halo (and even shorter as we tested with a 5-fold higher dye concentration). In the simulations, we neglected the short dye binding time and only included the time for co-translational folding. We estimated the co-translational folding time from the experiments with various C-terminal extensions, where we found that a length of $L_0 \sim 860$ nt provided enough time for HT proteins to start fluorescing when still bound to DNA (Fig. 1c). The reported elongation rate for transcription in *E. coli* cell lysate is around 10 nt s$^{-1}$[27], that is, we can estimate a co-translational folding time of ~86 s. In summary, after initiation of coupled transcription-translation on the gene, nascent HT peptides fold at the ribosome (or leave the gene for the shorter C-terminal extensions), react with a fluorogenic dye, and either bleach on the DNA molecule or are released from the DNA at the termination site after time $T$. To include additional experimental noise, e.g. camera read-out noise, we added uncorrelated Gaussian noise on the simulated intensity traces (Supplementary Fig. 6c).

By extracting the ACF from simulated intensity traces with the random production of nascent proteins and the parameters described above (Supplementary Fig. 6c), we found that the correlation time is shaped by the photostability of HT: At negligible bleaching rates, the ACF reproduced Eq. 3 with a linear decrease as function of $\tau$ reporting on transcription-translation initiation rates and gene length (Supplementary Fig. 6a, d, e). With dominating bleaching rates, the linear decay converges into an exponential decay (Supplementary Fig. 6d, e). The ACF is then better described with an exponential decay for the longer correlation times (Supplementary Fig. 6f). We extracted the

inverse amplitude $G(0)^{-1}$ and correlation time $\tau_C$ with a fit to the data with a mono-exponential decay and found that both quantities are reduced to the bleaching time of the reporter system (Supplementary Fig. 6g). As result, for longer genes that retain the HT protein longer on the gene, the apparent correlation time is reduced to the typical bleaching time of a single HT protein. However, in contrast to these simulated bounds because of bleaching (Supplementary Fig. 1d–h), we consistently found longer correlation times in our experimental data. The deviation suggested a time-correlated production of proteins from single DNA molecules. We concluded short bursts of transcription initiation events as discussed in the main text.

For the expression experiments with the antibiotic rifampicin, we considered an equilibrium binding between RNAP and rifampicin in solution

$$P + r \leftrightarrow Pr$$

with $P$ and $r$ as the concentration of *E. coli* RNAP and rifampicin, respectively. Solving the mass action law for steady-state conditions, we obtain

$$K_D = \frac{P\,r}{Pr} = \frac{(P_0 - Pr)(r_0 - Pr)}{Pr} \qquad (5)$$

with $P_0$ as initial level of RNAP, $r_0$ as the initial concentration of rifampicin, and $K_D$ as affinity constant. Solving the equation for the fraction of bound RNAP and rifampicin, we get

$$f = \frac{Pr}{P_0} = \frac{P_0 + r_0 + K_D - \sqrt{(P_0 + r_0 + K_D)^2 - 4P_0 r_0}}{2P_0} \qquad (6)$$

To fit Eq. 5 to our experimental data, we added amplitude and baseline to include the shift for the model-specific quantities of initiation rate and correlation time. Together, $G(0)^{-1}$ showed a decrease with rifampicin following Eq. 5 with literature-based values for $P_0 = 70\,nM$ and $K_D = 100\,nM$ (Fig. 3g)[27,62].

### Pulse detection in HT-T7 RNAP expression traces from full gene circuits

For the full genetic circuits with dCro-GFP and mCro-GFP, each HT-T7 RNAP trace was analyzed using the Fast-Fourier-Transformation (FFT) and the linear drift. The linear drift $D$ of the protein signal $I$ over time $t$ was computed with a fit to $I(t) = D \cdot t + b$, where $b$ is the baseline. The FFT amplitude and power spectrum were computed using $|FFT(I(t))|$ and $|FFT(I(t))|^2$, respectively (NumPy v1.20.3). The maximal FFT amplitude $\mathcal{F}_{max}$ was found and plotted against $D$ (Fig. 5d).

The ($D$, $\mathcal{F}_{max}$) values for expression traces from DNA molecules encoding the full circuit with mCro-GFP yielded the distribution $X_M$. We first assumed and then verified no/little negative feedback with this full circuit in bulk experiments (see Supplementary Fig. 9c), providing a reference distribution for the distribution $X_D$ obtained from DNA molecules with strong negative feedback through dCro-GFP. The pulsatile HT-T7 RNAP traces in both distributions $X_M$ and $X_D$ were then automatically classified by estimating the covariance matrix $Cov(X_M)$ and satisfying the condition $3 < (\mu_M - X_i)Cov(X_M)^{-1}(\mu_M - X_i)^T$, where $\mu_M$ is the average values of $X_M$.

### Statistics and reproducibility

No statistical method was used to predetermine sample size, but was on par with standard biophysical measurements. In titration experiments, mean values with less than 3 replicates were supported by adjacent mean values. No data was excluded from the analyses. As standard procedure in our field of study, the experiments were not randomized and the investigators were not blinded to allocation during experiments and outcome assessment.

## Reporting summary

Further information on research design is available in the Nature Portfolio Reporting Summary linked to this article.

## Data availability

The central microscopy data generated in this study have been deposited in the Zenodo database under accession code 8359540 (https://doi.org/10.5281/zenodo.8359540). Other relevant data are available within the paper as Source Data and Supplementary Information. Source data are provided with this paper.

## Code availability

Available at https://doi.org/10.5281/zenodo.10370923.

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

## Acknowledgements

We thank the Nanofabrication unit at the Weizmann Institute for support in the manufacturing process, the Forchheimer plasmid collection for the bacterial strain and plasmids, O. Vonshak and Y. Divon for providing the *gp* genes, D. Garenne for helping prepare the cell lysate, and H. Hofmann and A. Dupin for critically reading the manuscript. We thank J. Kumar for many fruitful discussions and finally, we thank J. Götz for her encouragement in this study. F.G. would like to thank EMBO (ALTF 598-2017) and Feinberg Graduate School for financial support with a long-term postdoctoral fellowship. We acknowledge funding from the Israel Science Foundation (R.B.Z. and S.S.D., grant no. 2723/19), the United States—Israel Binational Science Foundation (R.B.Z. and V.N. grant no. 2018208), the Isak Ferdinand and Dwosia Artmann Research Fund for Biological Physics, the Human Frontier Science Program (V.N., grant no. RGP0037/2015), and the Minerva Foundation (R.B.Z. and S.S.D., grant no. 712274). N.L. acknowledges the funding of the Max Planck Society. The research of N.L. was conducted within the Max Planck School Matter to Life supported by the German Federal Ministry of Education and Research (BMBF) in collaboration with the Max Planck Society. Y.B. is the incumbent of the Beatrice Barton Research Fellowship. E.W. acknowledges financial support from the German-Israeli Foundation for Scientific Research and Development (I–1196–195.9/2012).

## Author contributions

F.G. conceived the project, designed and performed experiments, and conceptualized the study. V.N. produced and helped with the cell-free expression system. Y.B. helped in constructing the bacterial strains. N.L. synthesized the fluorogenic dye. L.S. and E.W. produced and helped with the methyltransferase assay. F.G., R.B.Z. and S.S.D. discussed the results and wrote the manuscript with comments from all authors.

## Competing interests

The authors declare no competing interests.
