## [Peer review file · Nature Communications]

A genetic circuit on a single DNA molecule as an autonomous dissipative nanodeviceEditorial Note: This manuscript has been previously reviewed at another journal that is not operating a transparent peer review scheme. This document only contains reviewer comments and rebuttal letters for versions considered at Nature Communications.

REVIEWERS' COMMENTS

Reviewer #1 (Remarks to the Author):

Greiss et al describe the single-molecule analysis of coupled transcription-translation events and linked it to a synthetic regulatory circuit operating on a single DNA molecule. They improved their manuscript by performing additional experiments including cascades and feedback regulation for pulsatile dynamics. Also, the presentation of materials was improved for better readability. Thus, I would like to recommend the publication of manuscript.

There are a few minor errors that can be corrected.

Page 2, line 51: RNA polymerase (RNAP) -> RNAP (the abbreviation is already introduced n line 43)

Page 5, line 178: HT-T7-RNAP -> HT-T7 RNAP

Page 6, line 198: then -> than

References: article numbers are missing for some references e.g., ref 33, ref 43, ref 46.

Reviewer #3 (Remarks to the Author):

The authors have done a great job addressing our comments. This represents several important advances that will pave the way for routine applications of self-encoded dissipative nanodevices in the near future.

The work is now essentially ready for publication. We have only two small comments/questions to be considered in preparation of the final version.

We will find the term dissipative device somewhat confusing, but the authors have clearly explained the meaning they intend. They also provided a reference supporting this as an established terminology in their field. So we understand the decision to retain this term in the title.

In response to our comments, the authors included n values in several place throughout the manuscript. Do these n values represent biological replicates or two observations used for calculation of SD. We assume these are biological replicates, but it would be helpful if this was clarified. How many observations were used to calculate error and how many biological replicates were there. Ideally, both should be stated clearly. For example, in Figure 3 "Experiments were replicated for $n \geq 2$ (small circles) to obtain the average values (large circles)." To be clear, we are convinced by the results. This is only a small point for clarify whatever the explanation.

Reviewer #4 (Remarks to the Author):

The new manuscript incorporates a number of controls and additional experiments that makes it more convincing. The circuit presented in new fig5 is intriguing, but the intricate analysis makes it a bit confuse. Would it be possible to include an additional movie with mCro constructs for direct comparison?

I am still not entirely convinced by the contextualisation, but I believe that the technical achievements, and observation of coupled transcription/translation/regulation is highly significant. I thus recommend publication after checking the minor points below.

Minor comments:

"DNA nanotechnology seeks to construct nanodevices that perform mechanical work such as rotors and tweezers (1–3), or process information in the form of logic gates (4)". There are alternative way of processing information, e.g. neural architectures, Which are a current focus on DNA nanotech. ^[1]_{SEP}

"Dissipative DNA nanotechnology marks a shift to computing systems that could reversibly transition between states through a constant turnover of DNA and RNA strands, leading to pulsatile and oscillatory behavior in bulk, averaged over a large number of molecules (5, 6)." I understand the effort of the authors to try and classify "in solution" reaction networks (driven by concentration-controlled kinetics), and single-molecule devices (which require a different treatment) but the sentence is not completely clear. Additionally there is a number of molecular computing schemes that do not imply transcription, so DNA and/or RNA would be more appropriate.

"If signals in the protein time traces were the result of a random birth of a single protein, then $G(0)-1$ and typical timescale τ_0 should be bound by the bleaching time of around 1.5 mins and should not depend on the extension length"

than => then? $\frac{[I]}{[SEP]}$

(Supplementary Fig. 8b): Are the gene concentrations in bulk 1 nM as reported in Methods? Why is this simple cascade so inefficient?

"then all other cases" => than?

typo: "with the signals for HT-T7 RNAP and dCro-GFP (or mCro-GFP) signal "

REVIEWERS' COMMENTS

We again thank all three reviewers for their positive feedback and helpful comments on our updated manuscript. We responded to every comment and include our point-by-point response below (in blue) and marked the corresponding changes in the manuscript in red.

Reviewer #1 (Remarks to the Author):

Greiss et al describe the single-molecule analysis of coupled transcription-translation events and linked it to a synthetic regulatory circuit operating on a single DNA molecule. They improved their manuscript by performing additional experiments including cascades and feedback regulation for pulsatile dynamics. Also, the presentation of materials was improved for better readability. Thus, I would like to recommend the publication of manuscript.

There are a few minor errors that can be corrected.

Page 2, line 51: RNA polymerase (RNAP) -> RNAP (the abbreviation is already introduced n line 43)

Thank you for noting. We fixed the error.

Page 5, line 178: HT-T7-RNAP -> HT-T7 RNAP

Fixed.

Page 6, line 198: then -> than

Fixed.

References: article numbers are missing for some references e.g., ref 33, ref 43, ref 46.

Fixed.

Reviewer #3 (Remarks to the Author):

The authors have done a great job addressing our comments. This represents several important advances that will pave the way for routine applications of self-encoded dissipative nanodevices in the near future.

The work is now essentially ready for publication. We have only two small comments/questions to be considered in preparation of the final version.

We will find the term dissipative device somewhat confusing, but the authors have clearly explained the meaning they intend. They also provided a reference supporting this as an established terminology in their field. So we understand the decision to retain this term in the title.

Thank you.

In response to our comments, the authors included n values in several places throughout the manuscript. Do these n values represent biological replicates or two observations used for calculation of SD. We assume these are biological replicates, but it would be helpful if this was clarified. How many observations were used to calculate error and how many biological replicates were there. Ideally, both should be stated clearly. For example, in Figure 3 "Experiments were replicated for $n \geq 2$ (small circles) to obtain the average values (large circles)." To be clear, we are convinced by the results. This is only a small point for clarify whatever the explanation.

We added the information that these experiments were replicated independently and now explicitly write how many biological replicates were performed for each mean value.

Reviewer #4 (Remarks to the Author):

The new manuscript incorporates a number of controls and additional experiments that makes it more convincing. The circuit presented in new fig5 is intriguing, but the intricate analysis makes it a bit confuse. Would it be possible to include an additional movie with mCro constructs for direct comparison?

We gladly added another movie as Supplementary Movie 2 showing the protein synthesis dynamics of five DNA molecules encoding the full circuit with mCro.

I am still not entirely convinced by the contextualisation, but I believe that the technical achievements, and observation of coupled transcription/translation/regulation is highly significant. I thus recommend publication after checking the minor points below.

We thank this reviewer for the positive comments.

Minor comments:

"DNA nanotechnology seeks to construct nanodevices that perform mechanical work such as rotors and tweezers (1–3), or process information in the form of logic gates (4)". There are alternative way of processing information, e.g. neural architectures, Which are a current focus on DNA nanotech.

Thank you for your comment. We wanted to focus our manuscript on nanodevices, that is, single-molecule computing devices. Nevertheless, we added a reference and brief mentioning of neural architectures: "DNA nanotechnology seeks to construct nanodevices that perform mechanical work such as rotors and tweezers (1–3), or process information with architecture like logic gates (4) and neural networks (5, 6).

"Dissipative DNA nanotechnology marks a shift to computing systems that could reversibly transition between states through a constant turnover of DNA and RNA strands, leading to pulsatile and oscillatory behavior in bulk, averaged over a large number of molecules (5, 6)." I understand the effort of the authors to try and classify "in solution" reaction networks (driven by concentration-controlled kinetics), and single-molecule devices (which require a different treatment) but the sentence is not completely clear.

We clarified the statement: "Dissipative DNA nanotechnology marks a shift to computing systems that could reversibly transition between states through a constant turnover of DNA or RNA strands, leading to pulsatile and oscillatory behavior in bulk solution (7, 8)."

Additionally there is a number of molecular computing schemes that do not imply transcription, so DNA and/or RNA would be more appropriate.

Thank you for the comment. We modified the text accordingly.

"If signals in the protein time traces were the result of a random birth of a single protein, than $G(0)-1$ and typical timescale τ_0 should be bound by the bleaching time of around 1.5 mins and should not depend on the extension length"
than => then?

Fixed.

(Supplementary Fig. 8b): Are the gene concentrations in bulk 1 nM as reported in Methods? Why is this simple cascade so inefficient?

Indeed, the bulk gene concentrations were always kept at 1 nM. That is a total DNA concentration of 2 nM for the intermolecular case (split between two DNA molecules). We emphasize it now in the Methods section.

With a faster GFP production upon increasing HT-T7 RNAP, we concluded that HT-T7 RNAP could find the T7 promoter regulating the downstream *gfp* gene more easily in the intramolecular case (both genes encoded on a single DNA molecule).

We do not know why the intermolecular case is so inefficient compared to the intramolecular case. In bulk solution, a possible explanation is that the system runs out of energy before the produced HT-T7 RNAP finds the T7 promoter on the other DNA molecule to produce GFP.

"then all other cases" => than?

Fixed.

typo: "with the signals for HT-T7 RNAP and dCro-GFP (or mCro-GFP) signal "

Fixed.